# Needle Threading: Can LLMs Follow Threads through Near-Million-Scale Haystacks?

**Jonathan Roberts**[♦]    **Kai Han**[♠]    **Samuel Albanie**

[♦]University of Cambridge    [♠]The University of Hong Kong

https://needle-threading.github.io/

## Abstract

As the context limits of Large Language Models (LLMs) increase, the range of possible applications and downstream functions broadens. In many real-world tasks, decisions depend on details scattered across collections of often disparate documents containing mostly irrelevant information. Long-context LLMs appear well-suited to this form of complex information retrieval and reasoning, which has traditionally proven costly and time-consuming. However, although the development of longer context models has seen rapid gains in recent years, our understanding of how *effectively* LLMs *use their context* has not kept pace. To address this, we conduct a set of retrieval experiments designed to evaluate the capabilities of 17 leading LLMs, such as their ability to follow threads of information through the context window. Strikingly, we find that many models are remarkably *thread-safe*: capable of simultaneously following multiple threads without significant loss in performance. Still, for many models, we find the *effective context limit* is significantly shorter than the supported context length, with accuracy decreasing as the context window grows. Our study also highlights the important point that token counts from different tokenizers should not be directly compared—they often correspond to substantially different numbers of written characters. We release our code and long context experimental data.

## 1 Introduction

In recent years, LLMs and multimodal LLMs have been shown to possess remarkable capabilities (Bubeck et al., 2023) across tasks including software engineering (Hou et al., 2023), geospatial reasoning (Roberts et al., 2023a;b), medicine (Wu et al., 2023), mathematical and scientific figure understanding (Yue et al., 2024) and finance (Liu et al., 2023b). An expansion of compute resources, coupled with technical innovations (Liu et al., 2023a), is enabling contemporary frontier models to be trained on ever increasing volumes of data and longer *context limits*—the maximum number of tokens they can process at once. To contextualise the number of tokens leading models can process simultaneously, at just over 300k tokens[1], the classic novel *Moby-Dick* (Melville, 1851) could fit into the reported 2M token context window of Gemini 1.5 Pro (Reid et al., 2024) almost 5 times. As shown in Fig. 1, most books and even book series contain fewer tokens than the longest model context windows.

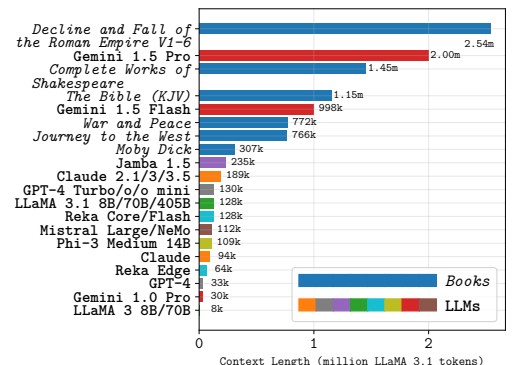

Figure 1: Contextualising context lengths of LLMs and classic literature[1]. Books sourced from Project Gutenberg (2024).

---

[1]Using the LLaMA-3.1 tokenizer (Dubey et al., 2024).

Emails: jdr53@cam.ac.uk, kaihanx@hku.hk, samuel.albanie.academic@gmail.com

A longer context offers potential benefits to performance, for example, many-shot in-context learning (Agarwal et al., 2024) in which hundreds or thousands of examples are appended to the model input. Another consequence is the wider range of possible applications and attainable downstream tasks. In particular, with a longer context, models can better perform real-world scenarios, such as legal document retrieval, academic research, understanding tax frameworks, and solving crimes and puzzles. In these cases, decisions are made and conclusions drawn based on large quantities of information distributed across many sources and formats. The ability to hold information – on the scale of multiple full-length novels or hundreds of academic papers and documents – in-context, makes models well-suited to this type of task.

The rate of development of longer context models has outpaced the understanding of how well they use their long context and can navigate it. Moreover, current benchmarks are considered inadequate and lacking (Bai et al., 2023; Zhang et al., 2024). Specifically, we identify three limitations of the extant literature related to long context understanding. **(1) Performance saturation**: Building on the 'needle in a haystack' test (Kamradt, 2023), numerous benchmarks focus on simple retrieval-based experiments. Frontier models can perform these tasks excellently, achieving perfect or near-perfect scores (Reid et al., 2024; Anthropic, 2024a; Dubey et al., 2024), leaving little headroom and useful insights to be gained. **(2) Limited context length**: In most long-context benchmarks, evaluations are limited to sub-100k contexts, falling short of the context limit of frontier LLMs by an order of magnitude. **(3) Lack of granular takeaways**: Due to the use of real documents or tendency to aggregate multiple tasks into an overall metric in most works, isolating specific trends is challenging other than the macro-trend that performance degrades as context length increases.

As such, there is opportunity for a set of challenging experiments, suitable to reach the limits of frontier models. To this end, we design and conduct a series of retrieval-based long context experiments of varying degrees of difficulty, across a range of context sizes up to 900k (Gemini 1.5) tokens. Our investigation includes novel *needle threading* tasks, which entail following a thread of linked pieces of information across different parts of the context and retrieving the final value. We also explore a more difficult *multi-threading* variation, which requires tracking multiple threads simultaneously, and assess whether the LLMs are thread-safe. We evaluate a suite of 17 LLMs on these tasks and observe performance decreases in longer contexts. Coupled with the finding that tokenization differs significantly between models, we introduce a task-specific effective context limit metric.

In summary, our core contributions are: (1) We introduce challenging multi-step *threading* and *multi-threading* retrieval tasks and evaluate 17 leading LLMs. (2) For simple needle retrieval tasks, we show that increased context length reduces performance, while increasing the number of needles retrieved concurrently has relatively limited impact on stronger models. (3) We show that leading LLMs are remarkably *thread-safe* - their thread following performance is largely unaffected by concurrent queries. (4) We compare tokenizers, highlighting significant differences in token counting. (5) We propose a task-specific and configurable model-agnostic effective context limit metric.

## 2 RELATED WORK

Evaluation of the long context capabilities of large language models is a recent yet burgeoning field of research. Numerous works focus on evaluating LLMs at long-document understanding tasks, such as question answering (An et al., 2023; Bai et al., 2023; Dong et al., 2023; Kuratov et al., 2024; Shaham et al., 2023; Li et al., 2023; Yuan et al., 2024), in which performance is generally found to decrease with increasing context length. Related tasks involve the summarisation and citation of insights across documents (Laban et al., 2024) and claim verification (Karpinska et al., 2024), which proves challenging for frontier models. While these benchmarks provide robust evaluations across a variety of tasks, they typically focus on smaller context lengths, with most including only limited explorations beyond 100k. Although there are benefits to realism by using real documents for these tasks, there are drawbacks. Specifically, timely annotation and curation are required, making it difficult to decompose performance as a function of variables such as context depth and length.

Other works focus on more abstract retrieval tasks (*e.g.*, Kamradt (2023)), allowing clearer takeaways at the cost of real-world relevance. An influential work is Liu et al. (2024), which empirically demonstrated that the position of relevant information within an LLM's context significantly impacts performance, with the best performances attained when information is at the beginning or end of the context. Similar behaviour is reported in some subsequent works (Xu et al., 2023; An et al., 2024;

Dong et al., 2023; Hsieh et al., 2024b; Laban et al., 2024) (and in some cases (Levy et al., 2024)) but others have failed to replicate the findings (Zhang et al., 2024; Song et al., 2024). Song et al. (2024) introduces a retrieval paradigm involving the accumulation of information throughout the context window, along with a more challenging variant that includes misleading information. Despite revealing interesting behaviour, there is limited headroom for frontier models on these tasks. Some recent related works include more challenging retrieval experiments, involving multiple steps. One example is the Ancestral Trace Challenge (Li et al., 2024), which proves challenging but is evaluated to relatively modest context lengths (up to 2k tokens). Another example is Variable Tracking (Hsieh et al., 2024a), however, results on these tasks are included as part of a wider set of experiments rather than being analysed in detail separately. We evaluate our difficult needle threading tasks to context lengths up to 630k tokens and comprehensively ablate and decompose the results.

## 3 TASKS

Taking inspiration from prior works (Liu et al., 2024; Hsieh et al., 2024a; Zhang et al., 2024), we focus our experimentation on abstract tasks containing synthetically generated data. By using synthetic data, (1) we avoid potentially expensive question-and-answer curation and annotation, (2) we ensure high-quality and noise-free data, and (3) we gain fine-grained control over the sequence length and other task parameters, allowing direct influence on difficulty. The abstract setting removes almost all natural language semantics, enabling the derivation of insights more closely linked to the parameters of the context window. We use string-serialised JSON objects containing key-value pairs of random UUIDs for our core experiments. Each UUID is a unique 32-character, 128-bit value string. The prompts used for each task follow this general structure:

> *<Task description>*
> {"9a159850-2f26-2bab-a114-4eefdeb0859f": "5de8eca9-8fd4-80b8-bf16-bd4397034f54",
> "d64b2470-8749-3be3-e6e8-11291f2dd06e": "1f22fcdb-9001-05ab-91f1-e7914b66a4ea",
> . . .,
> "bae328a1-44f3-7da1-d323-4bd9782beca1": "1183e29c-db7a-dccf-6ce8-c0a462d9942c",
> "5d88d112-e4ec-79a1-d038-8f1c58a240e4": "ea8bf5c3-1ede-7de0-ba05-d8cd69393423"}
> *<Output format instructions>*
> Key(s): "d64b2470-8749-3be3-e6e8-11291f2dd06e"
> Corresponding value(s):

In the following subsections, we outline our long-context understanding tasks. To complement the textual descriptions, we also include a schematic of each task in Fig 2. We conduct each experiment on a set of 'haystacks' of different sequence lengths, $m$, where each haystack ($H$) is a set of key-value pairs: $H = \{(K_i, V_i) \mid i \in \{1, 2, 3, ...m\}\}$.

**Single Needle.** In this simple, motivating task the goal is to provide the corresponding value ($V_i$) to a single specified key ($K_i$). For each haystack, we place needles at a fixed set of placement depths.

**Multiple Needles.** Building on the previous task, the goal of this task is to provide all the corresponding values to a specified set of between 2 and 25 keys. We consider two different placement methods: (1) *Random* - keys are randomly sampled (without replacement). (2) *Clustered* - after randomly sampling an initial key, all subsequent keys are sampled adjacently (motivated by the observation that informative cues for a given query often cluster together in real world applications).

**Conditional Needles.** Rather than providing specific keys, the goal of this task is to retrieve the values corresponding to all keys matching a specified criteria. In this case, we modify target keys by replacing a randomly selected character with a special character such as '*' or '&'. The expected values are those corresponding to keys containing the special character.

**Threading.** We define a *Threading Task* by initially selecting a subset of $n$ indices $j = \{j_1, j_2, ..., j_n\}$ from $H$, where $j_k \in \{1, 2, ..., m\}$. We then iterate over the indices $j$ for $k > 1$, replacing in $H$, $K_{j_k} \leftarrow V_{j_{k-1}}$, to form a thread. Given a single start key ($K_{j_1}$), the end goal is to find the value at the end of the thread ($V_{j_n}$). We evaluate thread lengths up to $n$=25 steps and experiment with different thread directions: (i) *Forward* - where the position of each subsequent pair in the thread occurs later in $H$ (*i.e.*, $j_1 < j_2 < ... < j_n$), (ii) *Backward* - where the positions of subsequent pairs occurs earlier in $H$ (*i.e.*, $j_1 > j_2 > ... > j_n$) and (iii) *Random* - where each subsequent pair in the thread can occur at any available position in $H$, regardless of direction.

**Multi-Threading.** For this task, we modify $H$ to include more than one thread. The goal is to determine the final value of *each thread*, given only the starting keys. We investigate different combinations of thread lengths, number of threads and thread direction.

**Branched Threading.** In this variation, we add branching to the threads. Specifically, at each index in the thread (except the first key), we modify 2 or more keys (number based on the specified branching factor, $b$) to equal one of the previous values. At each step, there are $b$ possible continuations, only one of which continues. The overall goal is to determine the final value of the longest thread.

## 4 EXPERIMENTS

**Baselines.** To build a comprehensive characterisation of the capabilities of current frontier long context models, we evaluated a set of 17 LLMs on our challenging long context retrieval experiments. Since the majority of frontier long context models are closed-source, we centre our evaluation on closed-source baselines. However, we also evaluate a subset of open-source models as a comparison. Where possible, we focus on chat or instruction-tuned variants of each LLM as their greater tendency to follow instructions enables a broader range of tasks and eases automatic evaluation. Specifically, we evaluate models from the closed-source GPT-4 (OpenAI, 2023; 2024), Gemini 1.0 (Gemini Team et al., 2023) and 1.5 (Reid et al., 2024), Claude 3 (Anthropic, 2024a) and 3.5 (Anthropic, 2024b),

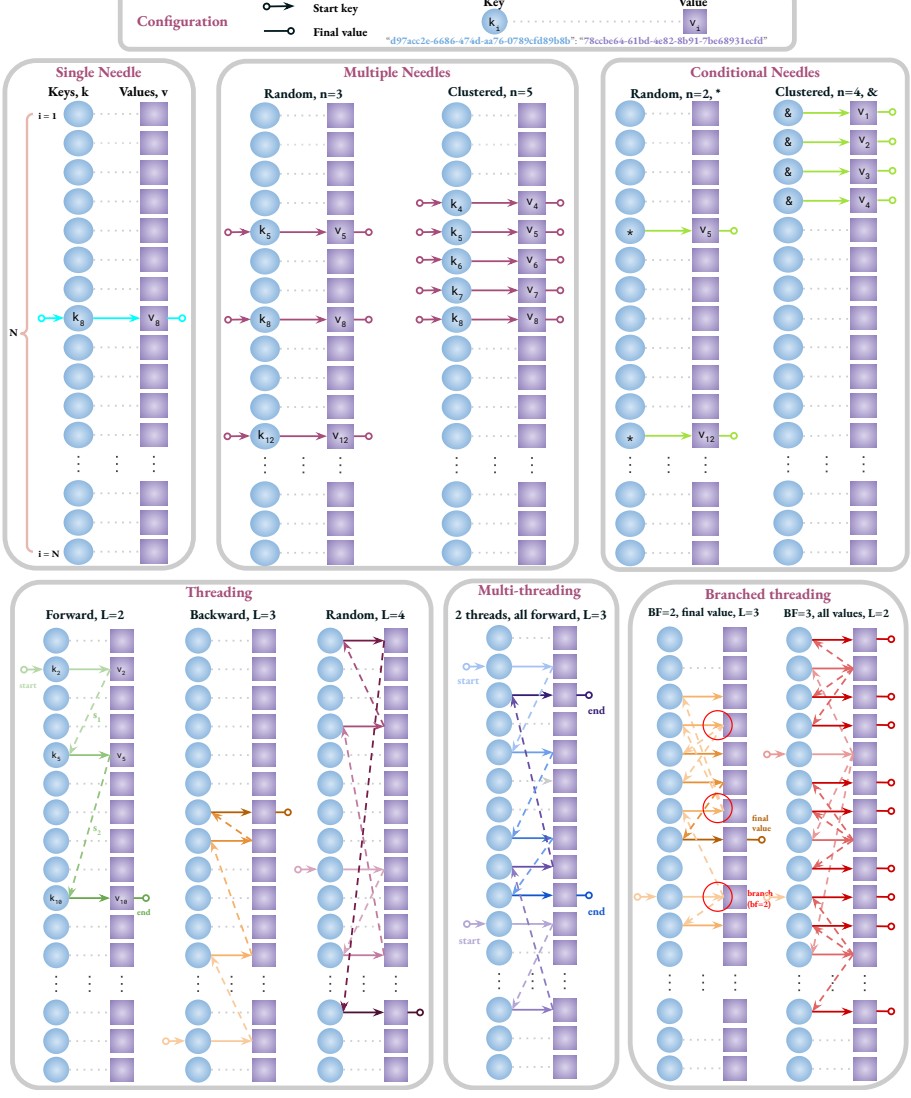

Figure 2: **Schematics for our long-context key-value retrieval tasks.** See §3 for descriptions.

and Reka (Ormazabal et al., 2024) series and the open-source Jamba 1.5 (Team et al., 2024), Mistral (AI, 2024a), and LLaMA 3.1 (Dubey et al., 2024) model series. Reported context lengths for each model are shown in Fig. 1.

**Prompting.** We used a simple prompting strategy throughout our experimentation that consisted of a single basic *user* prompt containing the question and output format instructions for each task. In keeping with prior works (Roberts et al., 2024a;b; OpenAI, 2024b), we do not modify the system prompt or tailor the prompt for each model. With the exception of providing examples of the desired output format, we do not use few-shot examples or explicitly encourage reasoning. We include the specific prompts used in each task in the .

**Inference.** All inference was carried out in a zero-shot setting. To aid reproducibility, we set model hyperparameters that encourage as deterministic generation as possible. Concretely, we use greedy search decoding strategies in which the most probable token is selected from the model vocabulary $V$ at each step, conditional on the preceding tokens *i.e.*, $w_{n+1} = \arg\max_{w \in V} P(w|w_1, w_2, \ldots, w_n)$. We achieve this by specifying random seeds and setting the $temperature$ parameter to zero. We evaluate the LLMs via the VertexAI (Google, 2024) {Gemini, Claude, Jamba, LLaMA 3.1, and Mistral}, OpenAI (OpenAI, 2024a) {GPT}, and Reka (AI, 2024b) {Reka} APIs. We aimed to evaluate each model as close to their context limits as possible, however, due to API restrictions this was not always feasible. More inference details can be found in the .

**Evaluation.** Following recent work (Roberts et al., 2024b), we use a strong LLM (Gemini 1.5 Flash) to parse the output from the evaluated LLMs into a specific format before evaluation via exact matching with the expected answer. As most models exhibit strong output following abilities, this LLM-based reformatting and evaluation has been demonstrated to correlate strongly with other evaluation measures in (Roberts et al., 2024a). For most models, this was only necessary for tasks requiring multiple values as the answer. For tasks requiring $k$ values as answers, we only evaluate the top $k$ answers provided by the models, any other additional answers were disregarded.

**Tokenization.** Context limits are typically reported in tokens and models are compared as though this is a consistent, model-agnostic metric. However, although minor variations in tokenization schemes might be expected across tokenizers, our preliminary experiments revealed significant differences, as outlined in Fig. 3. A UUID pair is represented by ~50 tokens by GPT-4o while Gemini 1.5 uses 75. Over longer contexts this difference is notable: Gemini 1.5 Flash's reported context limit of 1M tokens is equivalent to ~700k GPT-4o tokens. References to token counts throughout this section refer to text tokenized using the LLaMA 3.1 tokenizer.

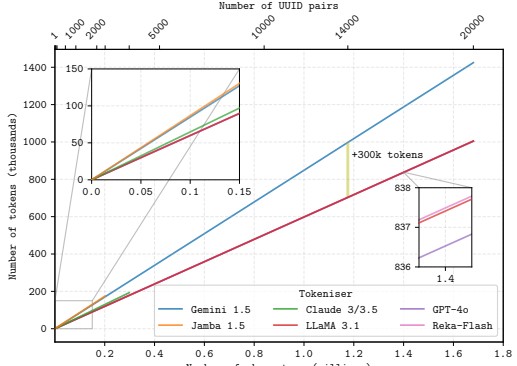

Figure 3: **Tokenization.** LLMs tokenize UUIDs at significantly different granularities.

In the following subsections, we report the results on the tasks outlined in §3. Experiments were carried out on haystacks of 12 different sizes ranging from 1k to 630k tokens (measured in LLaMA 3.1 tokens). For most models, we repeat each experiment on 5 different sets of haystacks and report the average performance, however, in some cases, only 1 repeat was feasible due to rate limit restrictions. More details, full results, and branched threading results can be found in the .

## 4.1 SINGLE NEEDLE

As a motivating task, we evaluate the ability of the models to accurately retrieve values corresponding to keys at fixed depths in 10% increments in the haystacks. We show heatmaps for a subset of the models in Fig. 4 and overall depth-averaged model performance on this task in the . At shorter contexts, the models perform this simple task well. However, in most cases, the **retrieval accuracy decreases for longer context lengths**. This suggests that while the models can perform inference on inputs up to their context limits, most have a smaller 'effective' limit from which they

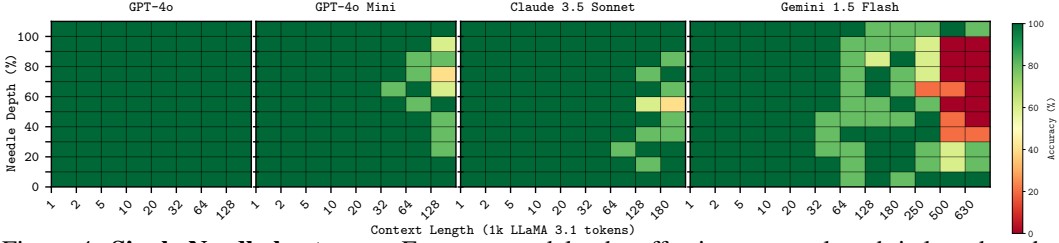

Figure 4: **Single Needle heatmaps**. For most models, the effective context length is less than the context limit. At longer contexts, retrieval precision decreases towards the middle of the context.

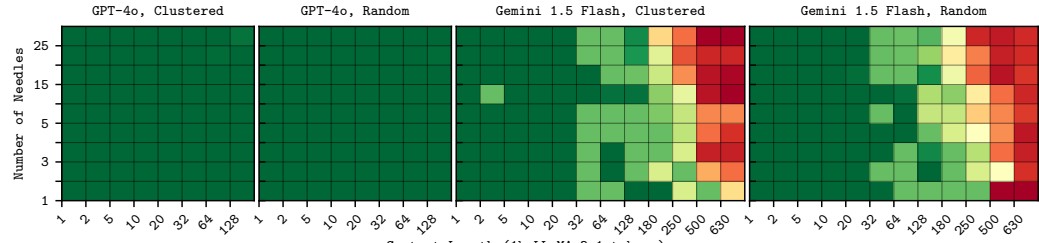

Figure 5: **Multiple Needles heatmaps**. Context length has a substantially greater effect on performance than needle placement positions or the number of needles.

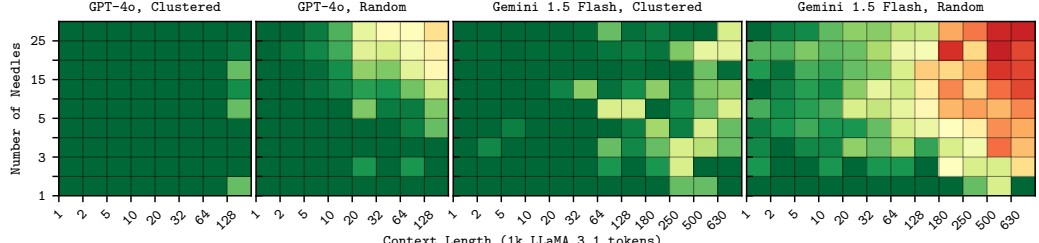

Figure 6: **Conditional Needles heatmaps**. Needles prove easier to retrieve when clustered.

can accurately extract information. Notable exceptions are GPT-4o and Jamba-1.5 Large, which attain perfect scores throughout. From the heatmaps, it is apparent that for the majority of models, **accuracy decreases towards the middle of the context**, supporting the findings of Liu et al. (2024).

## 4.2 MULTIPLE NEEDLES

Building on the previous task, we evaluate the capability to simultaneously retrieve values corresponding to [1,2,3,4,5,10,15,20,25] input keys from the haystacks. We report overall results averaged over all numbers of needles for each context size in Fig. 7 and heatmaps for selected models in Fig. 5, which show a decomposition of performance as a function of the number of needles and needle placement (randomly placed or clustered). Considering the overall result, we observe a similar macro-average trend as in the single needle task, where performance decreases at larger context sizes. However, in this case, owing to the higher degree of difficulty the performance drop-off is steeper, with several models' accuracy reduced to below 20% as their context limits are approached. This faster performance degradation suggests the effective context limits for this task are even shorter than when retrieving a single needle. As before, GPT-4o achieves a near-perfect score. The heatmaps for Gemini 1.5 Flash show **retrieval accuracy is unaffected by the relative placement of the needles**. Furthermore, **context length has a far larger impact on performance than the number of needles which has very limited impact on performance for the stronger models**.

## 4.3 CONDITIONAL NEEDLES

Sharing a similar structure to the multiple needles tasks, the conditional needles task assesses the ability to retrieve the values corresponding to [1,2,3,4,5,10,15,20,25] *unspecified* input keys that meet the condition of containing the '*' character. Compared to the multiple needles task, a similar

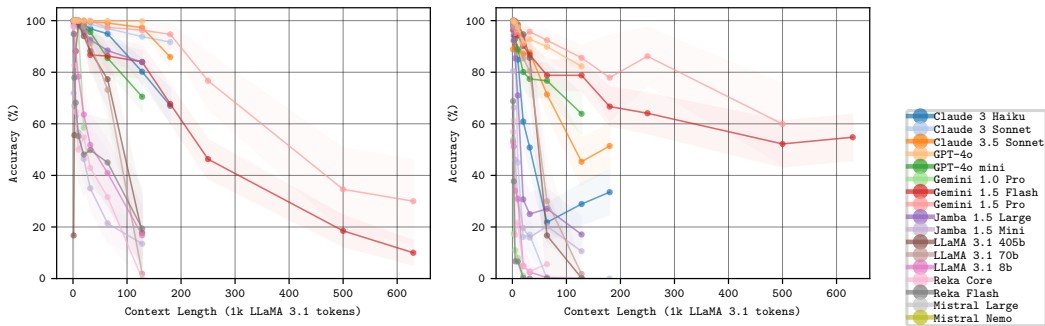

Figure 7: **Overall accuracy for Multiple Needles (left) and Conditional Needles (right).** Shaded regions show 95% confidence intervals.

overall trend is observed. Fig. 7 shows an arguably steeper initial performance decrease at shorter context lengths followed by a shallower decline towards the longer context lengths, resulting in lower overall scores. More differences between the tasks can be seen in the heatmaps in Fig. 6. One clear observation is that the placement of the conditional needles directly impacts the ability of the models to retrieve the corresponding values: **retrieval accuracy is higher when the relevant key-value pairs are clustered rather than randomly placed**. Also, when randomly placed, performance noticeably decreases when the number of needles increases. We found similar model performance with different conditional characters, though it was notably lower for '.'.

## 4.4 THREADING

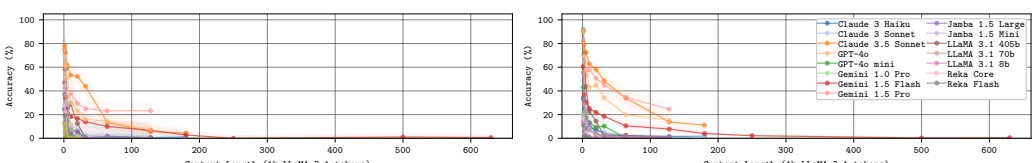

Figure 8: **Overall accuracy for Threading (left) and Multi-threading (right).** Shaded regions show 95% confidence intervals.

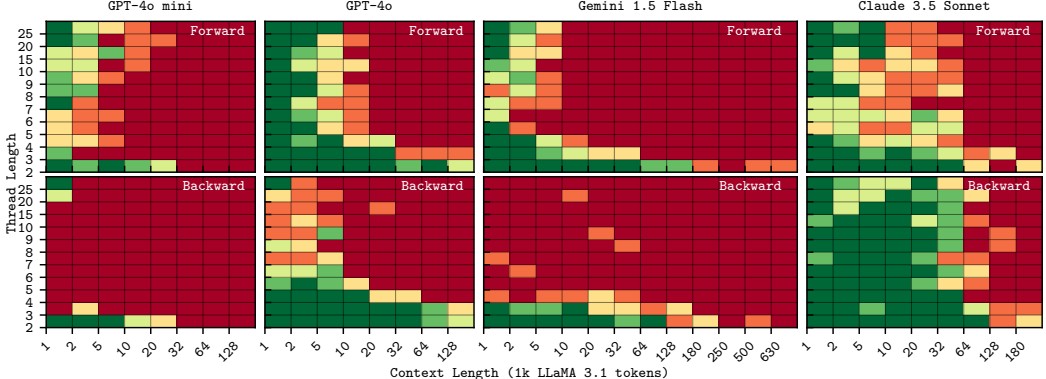

Figure 9: **Threading.** For most models, forward-travelling threads are easier to follow.

Having demonstrated the models' capabilities to perform single-step retrieval-based tasks (at least at shorter context lengths), we now move towards challenging multi-step reasoning-based retrieval. Concretely, at each context size, we test how accurately each model can retrieve the final value from threads of length: [2,3,4,5,6,7,8,9,10,15,20,25]. Threading introduces *directionality* – the relative position in the context window of subsequent pieces of the thread. We repeat each evaluation on threads going in forward, backward and random directions (see Fig. 2). Overall results are displayed in Fig. 8 and example heatmaps are shown in Fig. 9. Average accuracies are significantly lower for this task reflecting the added difficulty of following the thread through the context. For many models, *e.g.*, Gemini 1.5 Flash (darker red) and Claude 3 Haiku (darker blue), the accuracy plateaus to nearly zero at higher context lengths. The heatmaps reveal two clear trends. Firstly, **performance**

**decreases both with increasing context length and thread length**. Second, the direction of the thread matters. Except for Claude 3.5 Sonnet, **all models achieve much better accuracies on threads moving forward through the context compared to threads travelling backwards**.

## 4.5 MULTI-THREADING

We extend the threading task by adding extra threads for the models to simultaneously retrieve final values from. We evaluate on thread lengths of [2,3,4,5,10] for [2,3,4,5] separate threads and repeat for 'forwards', 'backwards', 'random directions', and 'all random' directions. The averaged accuracies for each context size are shown in Fig. 8. The lack of clear differences between the heatmaps for 2 vs 5 threads suggests that within the experimental range of thread lengths, **the models are thread-safe and performance is not significantly degraded by simultaneously following additional threads**. This is further illustrated in Fig. 10, in which Claude 3.5 Sonnet shows no performance degradation up to 25 threads and GPT-4o and Gemini 1.5 Pro show a gradual decline.

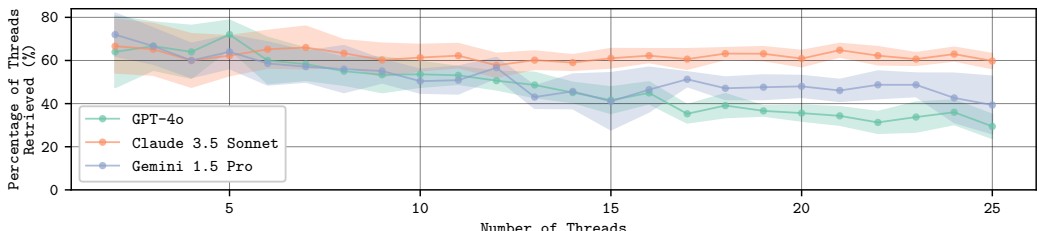

Figure 10: **Frontier LLMs are thread-safe.** Each point represents an average over 10 repeats retrieving randomly directed threads with a length of 3 in a 20k LLaMA 3.1 token haystack.

## 4.6 AGGREGATING HAYSTACK METRICS

To directly compare the overall performance of the models, we take an equally weighted average over the Single Needle, Multiple Needles, Conditional Needles, Threading and Multi-threading task scores. The results are presented in Tab. 1. We find that **the best model depends on the context size**: for the smallest contexts GPT-4o is best, at the longer contexts Gemini 1.5 Pro is superior, and Claude 3.5 Sonnet is the best performing from 2.5 to 32k. Across the board, the **closed-source models outperform the open-source models**.

| Model | Accuracy (%) | | | | | | | | | | | |
|---|---|---|---|---|---|---|---|---|---|---|---|---|
| | 1.2k | 2.5k | 5k | 10k | 20k | 32k | 64k | 128k | 180k | 250k | 500k | 630k |
| Gemini 1.5 Pro | 87.7 | 81.1 | 76.7 | 78.6 | 74.8 | 72.7 | **69.2** | **65.2** | - | - | - | - |
| Gemini 1.5 Flash | 80.7 | 73.3 | 70.1 | 67.5 | 65.7 | 60.1 | 53.9 | 53.3 | 46.1 | **37.4** | **21.3** | **19.7** |
| Jamba 1.5 Large | 70.8 | 63.5 | 60.2 | 57.5 | 47.1 | 43.9 | 43.4 | 40.4 | - | - | - | - |
| Jamba 1.5 Mini | 55.4 | 50.4 | 44.8 | 39.0 | 33.3 | 30.4 | 27.2 | 20.4 | - | - | - | - |
| Claude 3.5 Sonnet | 91.5 | **88.7** | **84.9** | **80.9** | **79.4** | **75.9** | 63.2 | 50.6 | **48.0** | - | - | - |
| Claude 3 Sonnet | 82.0 | 73.7 | 67.9 | 52.0 | 44.6 | 44.7 | 39.9 | 38.8 | 37.6 | - | - | - |
| Claude 3 Haiku | 71.8 | 65.7 | 62.8 | 59.3 | 53.3 | 50.3 | 43.0 | 37.2 | 37.4 | - | - | - |
| GPT-4o | **93.2** | 86.1 | 81.6 | 74.1 | 71.9 | 68.6 | 64.9 | 60.9 | - | - | - | - |
| GPT-4o mini | 75.7 | 67.9 | 64.7 | 61.8 | 58.3 | 56.3 | 51.3 | 42.9 | - | - | - | - |
| Reka Core | 59.8 | 53.8 | 17.0 | 33.5 | 29.6 | 27.0 | 24.9 | - | - | - | - | - |
| Reka Flash | 58.8 | 43.5 | 31.2 | 29.8 | 26.8 | 25.4 | 20.4 | 14.1 | - | - | - | - |
| LLaMA 3.1 8b | 54.9 | 49.8 | 45.3 | 40.9 | 33.6 | 29.0 | 26.0 | 13.7 | - | - | - | - |
| LLaMA 3.1 70b | 78.1 | 68.9 | 66.0 | 61.9 | 57.1 | 52.5 | 38.5 | 4.5 | - | - | - | - |
| LLaMA 3.1 405b | 76.7 | 77.1 | 70.5 | 69.8 | 62.8 | 55.2 | 39.3 | 19.6 | - | - | - | - |
| Gemini 1.0 Pro | 59.7 | 46.9 | 42.5 | 40.9 | 27.8 | - | - | - | - | - | - | - |

Table 1: **Overall results** averaged across the Single Needle, Multiple Needles, Conditional Needles, Threading and Multi-threading tasks. The highest scoring models at each context size is **bold**.

## 4.7 EFFECTIVE CONTEXT LENGTH

The observed macro-trend of reduced performance at longer context windows implies the models' ability to fully use their context window weakens as it grows. In short, there is a context size beyond which the models cannot effectively reason over and retrieve from. We propose an effective

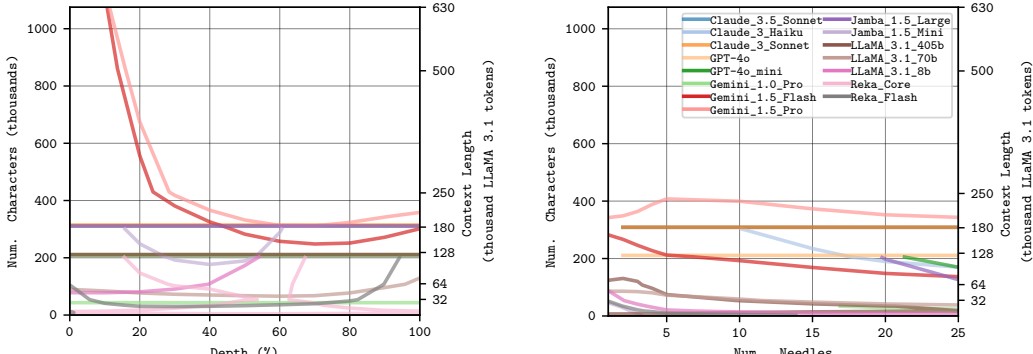

Figure 11: **Contour plots showing 'effective context length frontiers'** for the Single Needle (left) and Multiple Needles (right) tasks. Raw contours were used for the determination of the effective context lengths in Tab. 2. To improve visual clarity, the contours displayed have been smoothed using a Gaussian filter with $\sigma$=1.5.

| Model | Context Limit (1k chars) | Effective Context Size (1k chars) (proportion of limit, %) | | | | |
|---|---|---|---|---|---|---|
| | | Single Needle | Multiple Needles @10 needles | Conditional Needles @10 needles | Threading @5 steps | Multi-threading @5 steps |
| Gemini 1.5 Pro | 2472 | **315** (13%) | **430** (17%) | **220** (9%) | 0 (0%) | 0 (0%) |
| Gemini 1.5 Flash | 1236 | 132 (11%) | 294 (24%) | 44 (4%) | 0 (0%) | 0 (0%) |
| Jamba 1.5 Large | 295 | 295 (**100%**) | 295 (**100%**) | 10 (3%) | 0 (0%) | 0 (0%) |
| Jamba 1.5 Mini | 295 | 87 (29%) | 17 (6%) | 10 (3%) | 0 (0%) | 0 (0%) |
| Claude 3.5 Sonnet | 309 | 169 (55%) | 309 (**100%**) | 121 (39%) | 4 (1%) | **3** (**1%**) |
| Claude 3 Sonnet | 309 | 309 (**100%**) | 309 (**100%**) | 14 (5%) | 0 (0%) | 0 (0%) |
| Claude 3 Haiku | 309 | 87 (28%) | 201 (65%) | 18 (6%) | 0 (0%) | 0 (0%) |
| GPT-4o | 214 | 214 (**100%**) | 214 (**100%**) | 14 (7%) | **7** (**3%**) | **3** (**1%**) |
| GPT-4o mini | 214 | 120 (56%) | 176 (82%) | 43 (20%) | 0 (0%) | 0 (0%) |
| Reka Core | 214 | 5 (2%) | 5 (2%) | 3 (1%) | 0 (0%) | 0 (0%) |
| Reka Flash | 214 | 5 (2%) | 9 (4%) | 3 (1%) | 0 (0%) | 0 (0%) |
| LLaMA 3.1 8b | 214 | 14 (7%) | 22 (10%) | 34 (16%) | 0 (0%) | 0 (0%) |
| LLaMA 3.1 70b | 214 | 22 (10%) | 114 (53%) | 34 (16%) | 0 (0%) | 0 (0%) |
| LLaMA 3.1 405b | 214 | 138 (64%) | 124 (58%) | 60 (28%) | 0 (0%) | **3** (**1%**) |
| Gemini 1.0 Pro | 38 | 24 (63%) | 31 (82%) | 0 (0%) | 0 (0%) | 0 (0%) |

Table 2: **Effective context lengths.** @$X$ indicates the effective limit on the task when the named parameter equals $X$.

context length metric for each task that leverages the granularity of our experiments rather than simply estimating an average. For each task, we create a dense grid of points along the two key experimental variables (see axes of heatmaps) and interpolate the average accuracy at each point. We then determine a contour corresponding to a threshold accuracy level (taken here to be 75%). This contour represents the *effective frontier*, beyond which retrieval is unreliable. For the Single Needle task, we conservatively take the minimum value of the contour to provide a metric that is independent of context position. For the other tasks we take the corresponding contour value at a specific point on the x-axis, for example, where Num. Needles = 10 or Thread Length = 5. Example contour plots are shown in Fig. 11. Tab. 2 contains the computed effective context length metrics for each task. Given the discrepancies between tokenizers, we base our metric on the model-agnostic number of characters in the input rather than token count. The results show that **most models have an effective context length far less than their advertised context limit**.

## 4.8 NATURAL LANGUAGE ABLATION

To supplement the preceding experiments, we conduct natural language experiments that serve as closer analogues to real-world applications. Initially, we take sentences from *The History of the Decline and Fall of the Roman Empire*, by Edward Gibbon (see Fig.1) as a proxy for the UUID pairs in the abstract tasks. We prompt o1-preview (OpenAI, 2024) to generate a list of plausible yet fictional Roman events (*i.e.,* not included in the text). Using these events, we construct "threads" of linked sentences of the form '..., *Event A and then Event B.*',..., *'Event B and then Event C.'*,... and replace sentences in the text with them. We evaluate the threading task in this setting on haystacks from 1k to 630k token context lengths with threads of 2-25 steps (see Fig. 12). As in the abstract setting (Fig. 9), following threads in the natural language text proves challenging for the models, with similar poorer performance observed at longer contexts. The preference towards forward-travelling

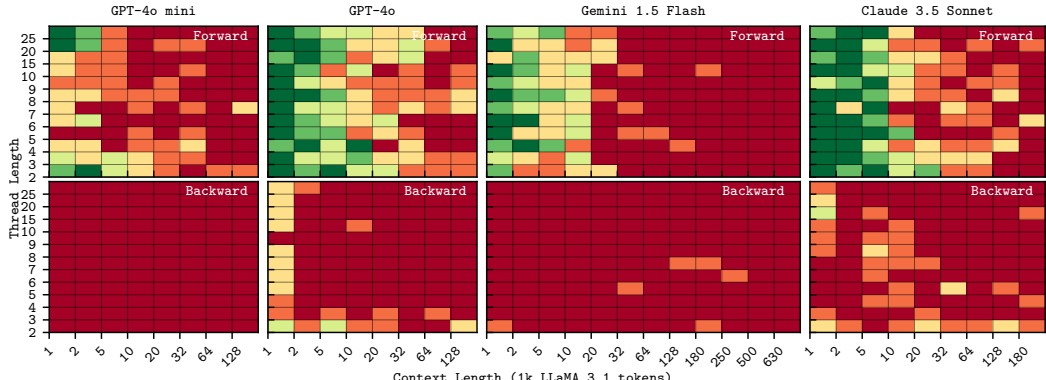

Figure 12: **Threading through natural text** showing a clear preference for forward moving threads.

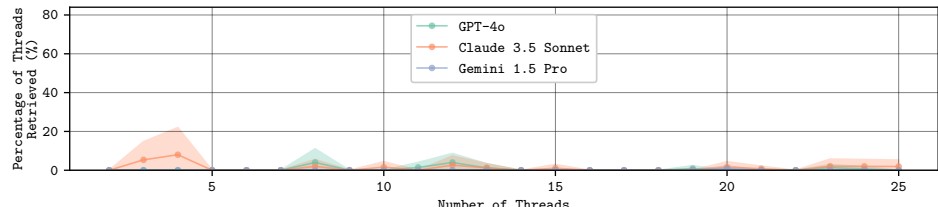

Figure 13: **Multi-threading through natural text.** Each point represents an average over 5 repeats retrieving **randomly directed threads** with a length of 3 in a ~20k LLaMA 3.1 token haystack.

threads is more apparent in this setting, with almost no backward-travelling threads correctly retrieved. We also conduct multi-threading experiments using this approach (this time with additional simultaneous threads) and present results in Fig. 13. Each point represents an average over 5 repeats retrieving randomly directed threads with a length of 3 in 20k LLaMA 3.1 token haystacks. Unlike the threading experiments – for which the results and insights are largely the same across the abstract and natural text settings – this multi-threading task in the natural language setting proved much more challenging for the models. Moreover, we find the task to be challenging when retrieving multiple threads that are all forward or all randomly directed. Thus, the multi-threading results are nuanced – with strong performance in the synthetic setting and weaker performance in the natural text setting.

# 5 CONCLUSIONS

We introduce a set of retrieval experiments covering simple single-needle retrieval, more difficult multiple-needle and conditional-needle retrieval and finally, challenging needle threading and multi-threading retrieval. All experiments are carried out on haystacks where the distractor text is from the same distribution as the relevant text. By curating the haystacks synthetically, we have granular control across specific independent variables enabling us to decompose key variables affecting performance and extract the following interesting takeaways after evaluating 17 LLMs on our tasks. (i) At long context lengths, the retrieval precision of frontier LLMs decreases towards the middle of the context; (ii) Clustering needles has little effect when tasked with retrieving specific needles but noticeably increases performance when retrieving all needles meeting a condition; (iii) Most LLMs achieve higher accuracies when retrieving threads moving forwards through the context versus backward directed threads; (iv) The evaluated LLMs show proficiency at keeping track of multiple threads simultaneously. Thus, we go further than most prior long context benchmarks, which provide only coarse, macro-trends. After revealing notable differences between tokenizers and observing poorer performances on larger haystacks, we derive an effective context limit metric. In particular, we propose a contour-based task-specific metric that is independent of tokenization. For a given task setting, the metric defines the maximum context size at which a model can effectively perform. We release our code and tasks for the community to use and we hope that our findings encourage further long context understanding research.

ACKNOWLEDGMENTS

This work was supported by the UKRI Centre for Doctoral Training in Application of Artificial Intelligence to the study of Environmental Risks (reference EP/S022961/1), an Isaac Newton Trust grant, a research gift from Google, an EPSRC HPC grant, the Hong Kong Research Grant Council - Early Career Scheme (Grant No. 27208022), and HKU Seed Fund for Basic Research. Samuel would like to acknowledge the support of Z. Novak and N. Novak in enabling his contribution.

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

APPENDIX

We structure our appendix into the following 8 parts:

A  Results for the Branched Threading task: §A.

B  Inference metrics such as API service response times: §B.

C  Details of the prompts used for each task: §C.

D  Specific API model versions used for inference: §D.

E  Full per-task results for each model: §E.

F  Discussion of the limitations of this work: §F

G  Description of API-based restrictions encountered during this work: §G.

H  Tables detailing the number of repeats carried out at different context lengths per model for each of the 5 core tasks: §H.

## A  BRANCHED THREADING

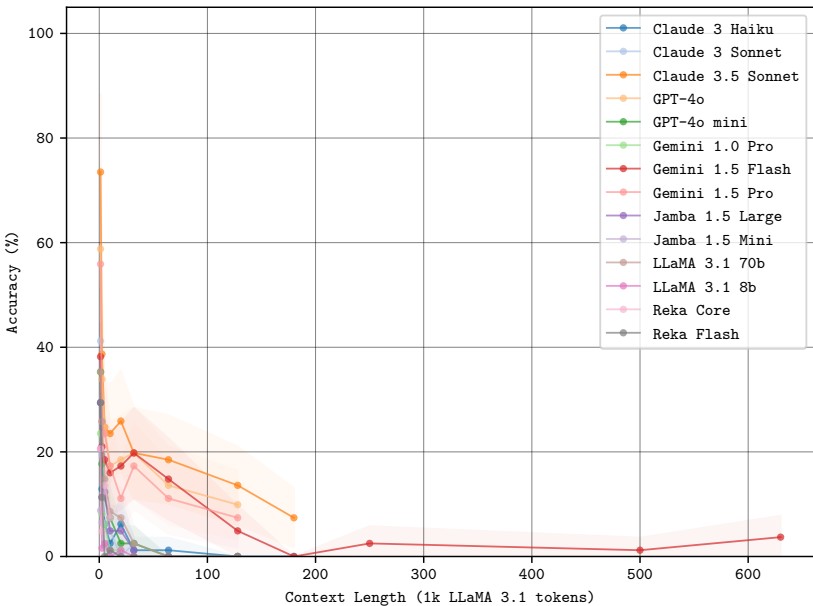

Figure 14: **Branched threading.** Shaded regions display 95% Wilson confidence intervals.

We carried out a branched threading investigation to evaluate the models' ability to accurately retrieve the final value of threads of length [2,3,4,5,6,7,8,9,10] where there is a branch at each step. We repeat this for branching factors of [2,3,4,5,6,7,8,9,10] and present the averaged results in Fig. 14. Similar to the threading tasks, retrieval accuracy drops significantly as the context length increases.

## B  INFERENCE METRICS

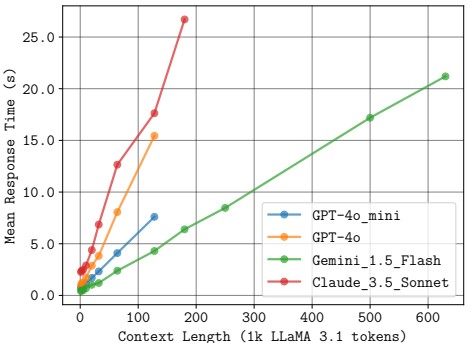
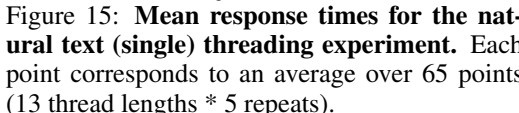
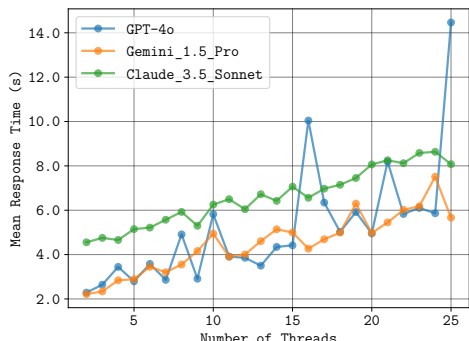

Figure 15: **Mean response times for the natural text (single) threading experiment.** Each point corresponds to an average over 65 points (13 thread lengths * 5 repeats).

Figure 16: **Mean response times for the natural text multi-threading experiment.** Each point corresponds to an average over 5 points (from 5 repeats).

## C PROMPTS

### C.1 SINGLE NEEDLE

Extract the value corresponding to the specified key in the JSON object below.
{
"9a159850-2f26-2bab-a114-4eefdeb0859f": "5de8eca9-8fd4-80b8-bf16-bd4397034f54",
"d64b2470-8749-3be3-e6e8-11291f2dd06e": "1f22fcdb-9001-05ab-91f1-e7914b66a4ea",
...,
"bae328a1-44f3-7da1-d323-4bd9782beca1": "1183e29c-db7a-dccf-6ce8-c0a462d9942c",
"5d88d112-e4ec-79a1-d038-8f1c58a240e4": "ea8bf5c3-1ede-7de0-ba05-d8cd69393423",
}
Only write the corresponding value, nothing else. Key: "<key>"
Corresponding value:

### C.2 MULTIPLE NEEDLES

Extract the values corresponding to the specified keys in the JSON object below.
{
"9a159850-2f26-2bab-a114-4eefdeb0859f": "5de8eca9-8fd4-80b8-bf16-bd4397034f54",
"d64b2470-8749-3be3-e6e8-11291f2dd06e": "1f22fcdb-9001-05ab-91f1-e7914b66a4ea",
...,
"bae328a1-44f3-7da1-d323-4bd9782beca1": "1183e29c-db7a-dccf-6ce8-c0a462d9942c",
"5d88d112-e4ec-79a1-d038-8f1c58a240e4": "ea8bf5c3-1ede-7de0-ba05-d8cd69393423",
}
Only write the list of corresponding values in square brackets, nothing else. Keys: [<keys>]
Corresponding values:

### C.3 CONDITIONAL NEEDLES

Extract the values corresponding to the keys that contain the character "<char>" in the JSON object below.
{
"9a159850-2f26-2bab-a114-4eefdeb0859f": "5de8eca9-8fd4-80b8-bf16-bd4397034f54",
"d64b2470-8749-3be3-e6e8-11291f2dd06e": "1f22fcdb-9001-05ab-91f1-e7914b66a4ea",
...,
"bae328a1-44f3-7da1-d323-4bd9782beca1": "1183e29c-db7a-dccf-6ce8-c0a462d9942c",
"5d88d112-e4ec-79a1-d038-8f1c58a240e4": "ea8bf5c3-1ede-7de0-ba05-d8cd69393423",
}
Only write the list of corresponding values in square brackets, nothing else.
Corresponding values:

## C.4 THREADING

The specified key corresponds to a value in the JSON object below. However, that value might equal another key in the JSON object. The value corresponding to this new key might also equal another key in the JSON object. This chain could continue beyond. Extract the final value in the chain. If the value corresponding to the first key does not equal another key, then the final value is the value corresponding to the first key.
{
"9a159850-2f26-2bab-a114-4eefdeb0859f": "5de8eca9-8fd4-80b8-bf16-bd4397034f54",
"d64b2470-8749-3be3-e6e8-11291f2dd06e": "1f22fcdb-9001-05ab-91f1-e7914b66a4ea",
. . .,
"bae328a1-44f3-7da1-d323-4bd9782beca1": "1183e29c-db7a-dccf-6ce8-c0a462d9942c",
"5d88d112-e4ec-79a1-d038-8f1c58a240e4": "ea8bf5c3-1ede-7de0-ba05-d8cd69393423",
}
Only write the corresponding value at the end of the chain, nothing else. Key: "<key>"
Corresponding final value:

## C.5 MULTI-THREADING

The specified keys each correspond to values in the JSON object below. However, the values might equal others key in the JSON object. The value corresponding to each new key might also equal another key in the JSON object. This chain could continue beyond. Extract the final values in each the chain. If the value corresponding to the first key does not equal another key, then the final value is the value corresponding to the first key.
{
"9a159850-2f26-2bab-a114-4eefdeb0859f": "5de8eca9-8fd4-80b8-bf16-bd4397034f54",
"d64b2470-8749-3be3-e6e8-11291f2dd06e": "1f22fcdb-9001-05ab-91f1-e7914b66a4ea",
. . .,
"bae328a1-44f3-7da1-d323-4bd9782beca1": "1183e29c-db7a-dccf-6ce8-c0a462d9942c",
"5d88d112-e4ec-79a1-d038-8f1c58a240e4": "ea8bf5c3-1ede-7de0-ba05-d8cd69393423",
}
Only write the corresponding values at the end of each chain in square brackets, nothing else. Keys: "<keys>"
Corresponding final values:

## C.6 BRANCHED THREADING

The specified key corresponds to a value in the JSON object below. However, that value might equal other keys in the JSON object. The values corresponding to these new keys might also equal other keys in the JSON object. This branched chain could continue beyond. Follow the longest chain and extract the final value at the end of the chain.
{
"9a159850-2f26-2bab-a114-4eefdeb0859f": "5de8eca9-8fd4-80b8-bf16-bd4397034f54",
"d64b2470-8749-3be3-e6e8-11291f2dd06e": "1f22fcdb-9001-05ab-91f1-e7914b66a4ea",
. . .,
"bae328a1-44f3-7da1-d323-4bd9782beca1": "1183e29c-db7a-dccf-6ce8-c0a462d9942c",
"5d88d112-e4ec-79a1-d038-8f1c58a240e4": "ea8bf5c3-1ede-7de0-ba05-d8cd69393423",
}
Only write the corresponding value at the end of the longest chain, nothing else. Key: "<key>"
Corresponding final value:

## C.7 LLM-REFORMATTING SINGLE VALUE OUTPUT

A generative model has answered a question to which the answer is a 32-character hexadecimal string UUID.\n The output from the model answering the question is "<unformatted_model_response>".\n Extract just the 32-character hexadecimal UUID string from the output. Keep the dashes but remove any whitespace, other characters (such as punctuation or quotes), and any additional text and explanation.\n Return only the extracted 32-character hexadecimal UUID, without any additional text or explanation. If no answer is provided, return "None".\n

## C.8 LLM-REFORMATTING MULTIPLE VALUE OUTPUT

A generative model has answered a question to which the answer is a list of 32-character hexadecimal strings.\n The output from the model answering the question is "<unformatted_model_response>".\n Extract just the list of 32-character hexadecimal UUID strings from the output. Keep the dashes but remove any whitespace, other characters (such as punctuation or quotes), and any additional text and explanation.\n Format the list as a list of strings, with each string in the list being a 32-character hexadecimal UUID string. For example: ['12345678-1234-5678-1234-567812345678', '87654321-4321-8765-4321-876587654321']\n Return only the extracted list, without any additional text or explanation. Do not include any additional syntax, like "'python"', in your answer. If no answer is provided, return "None".\n

## D  MODEL VERSIONS

Closed-source model API versions

- GPT-4o mini: *gpt-4o-mini-2024-07-18*
- GPT-4o: *gpt-4o-2024-08-06*
- Gemini-Pro: *gemini-1.0-pro-002*
- Gemini 1.5 Flash: *gemini-1.5-flash-preview-0514*
- Gemini 1.5 Pro: *gemini-1.5-pro-preview-0514*
- Claude 3 Haiku: *claude-3-haiku@20240307*
- Claude 3 Sonnet: *claude-3-sonnet@20240229*
- Claude 3.5 Sonnet: *claude-3-5-sonnet@20240620*
- Reka Flash: *reka-flash-20240904*
- Reka Core: *reka-core-20240415*

## E  FULL RESULTS

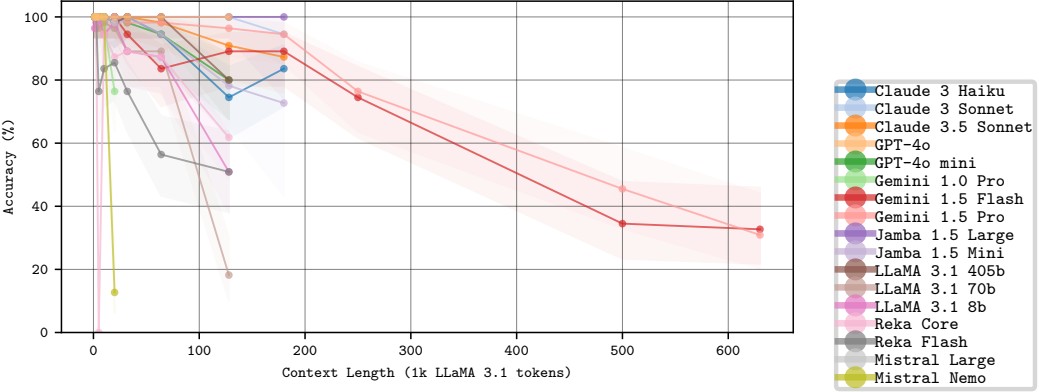

Figure 17: **Single Needle overall performance** with 95% Wilson confidence intervals.

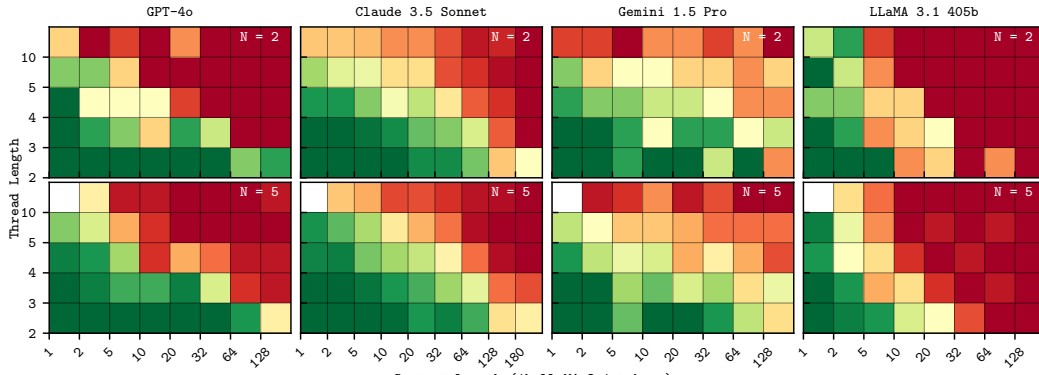

Figure 18: **Multi-threading.** Concurrently following $N$ threads does not degrade performance.

| Model | Accuracy (%) | | | | | | | | | | | |
|---|---|---|---|---|---|---|---|---|---|---|---|---|
| | 1.2k | 2.5k | 5k | 10k | 20k | 32k | 64k | 128k | 180k | 250k | 500k | 630k |
| Gemini 1.5 Pro | 100.0 | 100.0 | 100.0 | 100.0 | 100.0 | 98.2 | 98.2 | 96.4 | 94.5 | 76.4 | 45.5 | 30.9 |
| Gemini 1.5 Flash | 100.0 | 100.0 | 100.0 | 100.0 | 100.0 | 94.5 | 83.6 | 89.1 | 89.1 | 74.5 | 34.5 | 32.7 |
| Jamba 1.5 Large | 100.0 | 100.0 | 100.0 | 100.0 | 100.0 | 100.0 | 100.0 | 100.0 | 100.0 | - | - | - |
| Jamba 1.5 Mini | 100.0 | 100.0 | 98.2 | 98.2 | 96.4 | 100.0 | 94.5 | 78.2 | 72.7 | - | - | - |
| Claude 3.5 Sonnet | 100.0 | 100.0 | 100.0 | 100.0 | 100.0 | 100.0 | 98.2 | 90.9 | 87.3 | - | - | - |
| Claude 3 Sonnet | 100.0 | 100.0 | 100.0 | 100.0 | 100.0 | 100.0 | 100.0 | 100.0 | 94.5 | - | - | - |
| Claude 3 Haiku | 100.0 | 100.0 | 100.0 | 100.0 | 98.2 | 100.0 | 94.5 | 74.5 | 83.6 | - | - | - |
| GPT-4o | 100.0 | 100.0 | 100.0 | 100.0 | 100.0 | 100.0 | 100.0 | 100.0 | - | - | - | - |
| GPT-4o mini | 100.0 | 100.0 | 100.0 | 100.0 | 100.0 | 98.2 | 94.5 | 80.0 | - | - | - | - |
| Reka Core | 100.0 | 100.0 | 0.0 | 94.5 | 87.3 | 89.1 | 87.3 | 61.8 | - | - | - | - |
| Reka Flash | 100.0 | 100.0 | 76.4 | 83.6 | 85.5 | 76.4 | 56.4 | 50.9 | - | - | - | - |
| LLaMA 3.1 8b | 96.4 | 98.2 | 100.0 | 94.5 | 98.2 | 89.1 | 87.3 | 50.9 | - | - | - | - |
| LLaMA 3.1 70b | 100.0 | 96.4 | 96.4 | 98.2 | 96.4 | 89.1 | 89.1 | 18.2 | - | - | - | - |
| LLaMA 3.1 405b | 100.0 | 100.0 | 100.0 | 100.0 | 98.2 | 100.0 | 100.0 | 80.0 | - | - | - | - |
| Gemini 1.0 Pro | 100.0 | 100.0 | 100.0 | 98.2 | 76.4 | - | - | - | - | - | - | - |
| Mistral Large | 100.0 | 100.0 | 100.0 | 100.0 | 98.2 | - | - | - | - | - | - | - |
| Mistral Nemo | 100.0 | 100.0 | 100.0 | 100.0 | 12.7 | - | - | - | - | - | - | - |

Table 3: Single Needle depth-averaged results. Reka Core 0.0 at 5k is likely due to safety restraints (output is not generated due to 'context').

| Model | Accuracy (%) | | | | | | | | | | | |
|---|---|---|---|---|---|---|---|---|---|---|---|---|
| | 1.2k | 2.5k | 5k | 10k | 20k | 32k | 64k | 128k | 180k | 250k | 500k | 630k |
| Gemini 1.5 Pro | 100.0 | 100.0 | 100.0 | 100.0 | 100.0 | 99.8 | 97.4 | 96.3 | 94.7 | 76.7 | 34.6 | 30.0 |
| Gemini 1.5 Flash | 100.0 | 98.9 | 100.0 | 100.0 | 99.9 | 86.7 | 86.3 | 84.0 | 67.7 | 46.3 | 18.5 | 10.0 |
| Jamba 1.5 Large | 99.6 | 99.4 | 99.5 | 98.0 | 95.5 | 92.6 | 88.4 | 83.9 | - | - | - | - |
| Jamba 1.5 Mini | 71.9 | 67.0 | 63.0 | 56.6 | 46.4 | 35.0 | 21.4 | 13.5 | - | - | - | - |
| Claude 3.5 Sonnet | 100.0 | 100.0 | 100.0 | 99.9 | 99.7 | 99.6 | 99.1 | 97.3 | 85.9 | - | - | - |
| Claude 3 Sonnet | 100.0 | 100.0 | 100.0 | 100.0 | 99.5 | 98.6 | 97.0 | 93.8 | 91.7 | - | - | - |
| Claude 3 Haiku | 99.9 | 100.0 | 99.4 | 99.7 | 98.5 | 96.9 | 94.9 | 80.2 | 67.0 | - | - | - |
| GPT-4o | 100.0 | 100.0 | 100.0 | 100.0 | 100.0 | 100.0 | 99.9 | 99.8 | - | - | - | - |
| GPT-4o mini | 99.9 | 99.8 | 99.0 | 98.6 | 97.2 | 95.6 | 85.5 | 70.5 | - | - | - | - |
| Reka Core | 97.6 | 82.7 | 64.7 | 50.0 | 54.8 | 42.9 | 31.6 | 0.0 | - | - | - | - |
| Reka Flash | 94.9 | 77.9 | 68.2 | 55.2 | 48.1 | 49.8 | 45.0 | 19.4 | - | - | - | - |
| LLaMA 3.1 8b | 98.0 | 94.7 | 88.1 | 78.3 | 63.6 | 51.8 | 40.9 | 16.8 | - | - | - | - |
| LLaMA 3.1 70b | 100.0 | 100.0 | 100.0 | 99.9 | 97.7 | 91.2 | 73.2 | 1.9 | - | - | - | - |
| LLaMA 3.1 405b | 16.7 | 55.6 | 88.2 | 98.6 | 94.0 | 88.2 | 77.3 | 17.7 | - | - | - | - |
| Gemini 1.0 Pro | 99.8 | 99.9 | 98.2 | 97.4 | 58.5 | - | - | - | - | - | - | - |

Table 4: Multiple Needles overall results.

| Model | Accuracy (%) | | | | | | | | | | | |
|---|---|---|---|---|---|---|---|---|---|---|---|---|
| | 1.2k | 2.5k | 5k | 10k | 20k | 32k | 64k | 128k | 180k | 250k | 500k | 630k |
| Gemini 1.5 Pro | 98.6 | 98.3 | 95.2 | 97.3 | 93.6 | 95.7 | 92.4 | 85.6 | 77.9 | 86.2 | 59.9 | - |
| Gemini 1.5 Flash | 96.3 | 96.9 | 94.6 | 94.3 | 90.2 | 86.8 | 78.8 | 78.8 | 66.7 | 64.1 | 52.2 | 54.8 |
| Jamba 1.5 Large | 98.0 | 92.4 | 85.4 | 71.0 | 30.7 | 25.0 | 27.1 | 17.1 | - | - | - | - |
| Jamba 1.5 Mini | 80.5 | 66.3 | 46.0 | 30.7 | 19.6 | 15.9 | 20.3 | 10.6 | - | - | - | - |
| Claude 3.5 Sonnet | 88.9 | 92.2 | 89.8 | 88.3 | 87.1 | 87.7 | 71.4 | 45.3 | 51.4 | - | - | - |
| Claude 3 Sonnet | 99.9 | 99.9 | 98.1 | 45.0 | 16.1 | 17.0 | 0.0 | 0.1 | 0.0 | - | - | - |
| Claude 3 Haiku | 99.2 | 94.3 | 90.2 | 84.9 | 60.9 | 50.8 | 21.8 | 28.9 | 33.5 | - | - | - |
| GPT-4o | 100.0 | 99.8 | 99.2 | 97.5 | 91.2 | 92.8 | 89.9 | 82.3 | - | - | - | - |
| GPT-4o mini | 98.2 | 98.3 | 92.9 | 88.9 | 80.1 | 77.4 | 76.7 | 63.9 | - | - | - | - |
| Reka Core | 56.9 | 61.2 | 16.9 | 21.7 | 4.7 | 2.8 | 5.6 | - | - | - | - | - |
| Reka Flash | 68.8 | 37.7 | 6.7 | 6.6 | 0.2 | 0.0 | 0.0 | 0.0 | - | - | - | - |
| LLaMA 3.1 8b | 52.9 | 51.2 | 34.1 | 31.0 | 4.9 | 2.5 | 0.4 | 0.0 | - | - | - | - |
| LLaMA 3.1 70b | 97.2 | 98.4 | 99.1 | 97.1 | 85.4 | 80.5 | 30.0 | 1.8 | - | - | - | - |
| LLaMA 3.1 405b | 100.0 | 100.0 | 99.8 | 98.5 | 94.7 | 85.6 | 16.7 | 0.2 | - | - | - | - |
| Gemini 1.0 Pro | 54.0 | 17.4 | 11.0 | 8.0 | 1.1 | - | - | - | - | - | - | - |

Table 5: Conditional Needles overall results.

| Model | Accuracy (%) | | | | | | | | | | | |
|---|---|---|---|---|---|---|---|---|---|---|---|---|
| | 1.2k | 2.5k | 5k | 10k | 20k | 32k | 64k | 128k | 180k | 250k | 500k | 630k |
| Gemini 1.5 Pro | 57.8 | 42.2 | 35.0 | 37.8 | 29.4 | 25.0 | 23.3 | 23.3 | - | - | - | - |
| Gemini 1.5 Flash | 46.7 | 33.9 | 25.6 | 18.3 | 16.7 | 13.9 | 10.0 | 6.7 | 2.8 | 0.0 | 1.1 | 0.6 |
| Jamba 1.5 Large | 23.9 | 12.2 | 8.3 | 5.6 | 5.6 | 0.6 | 1.1 | 0.0 | - | - | - | - |
| Jamba 1.5 Mini | 5.6 | 7.8 | 3.3 | 1.7 | 1.7 | 0.0 | 0.0 | 0.0 | - | - | - | - |
| Claude 3.5 Sonnet | 78.3 | 72.2 | 61.7 | 53.3 | 52.2 | 43.9 | 13.3 | 5.6 | 4.4 | - | - | - |
| Claude 3 Sonnet | 40.0 | 26.7 | 17.2 | 7.2 | 6.7 | 2.8 | 1.1 | 0.0 | 0.0 | - | - | - |
| Claude 3 Haiku | 25.6 | 10.0 | 7.2 | 3.3 | 1.7 | 0.0 | 1.7 | 0.6 | 1.1 | - | - | - |
| GPT-4o | 75.0 | 61.1 | 51.1 | 30.0 | 23.3 | 16.1 | 14.4 | 7.2 | - | - | - | - |
| GPT-4o mini | 37.2 | 22.8 | 14.4 | 8.3 | 5.0 | 0.0 | 0.0 | 0.0 | - | - | - | - |
| Reka Core | 27.8 | 22.2 | 0.0 | 0.0 | 0.0 | 0.0 | 0.0 | - | - | - | - | - |
| Reka Flash | 19.4 | 0.0 | 2.8 | 2.8 | 0.0 | 0.0 | 0.0 | 0.0 | - | - | - | - |
| LLaMA 3.1 8b | 13.2 | 1.4 | 0.7 | 0.0 | 0.0 | 0.0 | 0.0 | 0.0 | - | - | - | - |
| LLaMA 3.1 70b | 38.0 | 21.3 | 13.0 | 7.4 | 1.9 | 0.0 | 0.0 | 0.0 | - | - | - | - |
| LLaMA 3.1 405b | 75.0 | 58.3 | 20.8 | 29.2 | 12.5 | 0.0 | 0.0 | 0.0 | - | - | - | - |
| Gemini 1.0 Pro | 23.3 | 8.9 | 2.2 | 0.6 | 1.1 | - | - | - | - | - | - | - |
| Mistral Large | 68.9 | 45.0 | 31.1 | 10.6 | 1.1 | - | - | - | - | - | - | - |
| Mistral Nemo | 12.2 | 7.2 | 2.2 | 0.0 | 0.0 | - | - | - | - | - | - | - |

Table 6: Threading overall results.

| Model | Accuracy (%) | | | | | | | | | | | |
|---|---|---|---|---|---|---|---|---|---|---|---|---|
| | 1.2k | 2.5k | 5k | 10k | 20k | 32k | 64k | 128k | 180k | 250k | 500k | 630k |
| Gemini 1.5 Pro | 82.2 | 65.1 | 53.2 | 57.9 | 50.7 | 44.9 | 34.6 | 24.6 | - | - | - | - |
| Gemini 1.5 Flash | 60.5 | 36.9 | 30.4 | 25.1 | 21.9 | 18.5 | 10.5 | 7.8 | 4.0 | 2.2 | 0.3 | 0.5 |
| Jamba 1.5 Large | 32.5 | 13.5 | 8.0 | 13.0 | 3.8 | 1.2 | 0.6 | 1.2 | - | - | - | - |
| Jamba 1.5 Mini | 18.9 | 10.8 | 13.6 | 7.9 | 2.5 | 1.0 | 0.0 | 0.0 | - | - | - | - |
| Claude 3.5 Sonnet | 90.1 | 79.1 | 72.8 | 62.8 | 58.2 | 48.5 | 33.9 | 13.8 | 11.1 | - | - | - |
| Claude 3 Sonnet | 69.9 | 42.1 | 24.2 | 7.6 | 1.0 | 5.1 | 1.5 | 0.0 | 1.6 | - | - | - |
| Claude 3 Haiku | 34.1 | 24.2 | 17.4 | 8.7 | 7.4 | 4.0 | 2.3 | 1.6 | 1.6 | - | - | - |
| GPT-4o | 90.9 | 69.5 | 57.5 | 42.9 | 44.9 | 34.1 | 19.9 | 15.2 | - | - | - | - |
| GPT-4o mini | 43.0 | 18.6 | 17.3 | 13.1 | 9.3 | 10.3 | 0.0 | 0.0 | - | - | - | - |
| Reka Core | 16.8 | 2.9 | 3.5 | 1.5 | 1.3 | 0.0 | 0.2 | - | - | - | - | - |
| Reka Flash | 11.1 | 1.7 | 2.0 | 0.7 | 0.2 | 0.6 | 0.8 | 0.0 | - | - | - | - |
| LLaMA 3.1 8b | 14.0 | 3.3 | 3.5 | 0.9 | 1.1 | 1.5 | 1.6 | 0.6 | - | - | - | - |
| LLaMA 3.1 70b | 55.1 | 28.3 | 21.6 | 6.7 | 4.1 | 1.8 | 0.3 | 0.4 | - | - | - | - |
| LLaMA 3.1 405b | 91.6 | 71.5 | 43.7 | 22.7 | 14.5 | 2.2 | 2.4 | 0.3 | - | - | - | - |
| Gemini 1.0 Pro | 21.6 | 8.2 | 1.3 | 0.3 | 1.9 | - | - | - | - | - | - | - |
| Mistral Large | 71.3 | 49.2 | 34.9 | 14.4 | 8.7 | - | - | - | - | - | - | - |
| Mistral Nemo | 19.0 | 14.4 | 9.7 | 7.7 | 3.1 | - | - | - | - | - | - | - |

Table 7: Multi-Threading overall results.

## F LIMITATIONS

We note several limitations to our work. First, we restrict our study to the use of synthetic data. While this has significant benefits (fine-grained controllability, automatic provision of perfect ground truth), our benchmark does not capture differences in LLM behaviour that are domain-specific (for instance, LLMs may be more performant on some distributions than others). Second, as discussed below, the scale of our experiments (particular the number of experimental repeats) was limited by cost for the larger models.

## G API RESTRICTIONS

The design of our experiments was guided in part by the following API-based restrictions and limitations:

- **Cost**. For the most expensive models (*e.g.*, Gemini 1.5 Pro, Claude 3.5 Sonnet), running just a single repeat on one task could cost hundreds of dollars. Therefore, in some cases, the evaluation of these models could not be repeated extensively, limiting the statistical strength of our experiments.
- **Context restrictions**. Some models were only available for API-based inference in a limited capacity (*e.g.*, Mistral), in which it was not possible to provide inputs that approach the context limit. As such, we could only evaluate these models as close to the context limit as we could.
- **Latency**. As a result of latency introduced by low server throughput or indirectly via low rate limits at the time of writing, for some models (*e.g.*, LLaMA 3.1), it was not possible to extensively conduct repeats.

## H REPEATS

| Model | Num. Repeats | | | | | | | | | | | |
|---|---|---|---|---|---|---|---|---|---|---|---|---|
| | 1.2k | 2.5k | 5k | 10k | 20k | 32k | 64k | 128k | 180k | 250k | 500k | 630k |
| Gemini 1.5 Pro | 5 | 5 | 5 | 5 | 5 | 5 | 5 | 5 | 5 | 5 | 5 | 5 |
| Gemini 1.5 Flash | 5 | 5 | 5 | 5 | 5 | 5 | 5 | 5 | 5 | 5 | 5 | 5 |
| Jamba 1.5 Large | 5 | 5 | 5 | 5 | 5 | 5 | 5 | 5 | 1 | - | - | - |
| Jamba 1.5 Mini | 5 | 5 | 5 | 5 | 5 | 5 | 5 | 5 | 1 | - | - | - |
| Claude 3.5 Sonnet | 5 | 5 | 5 | 5 | 5 | 5 | 5 | 5 | 5 | - | - | - |
| Claude 3 Sonnet | 5 | 5 | 5 | 5 | 5 | 5 | 5 | 5 | 5 | - | - | - |
| Claude 3 Haiku | 5 | 5 | 5 | 5 | 5 | 5 | 5 | 5 | 5 | - | - | - |
| GPT-4o | 5 | 5 | 5 | 5 | 5 | 5 | 5 | 5 | - | - | - | - |
| GPT-4o mini | 5 | 5 | 5 | 5 | 5 | 5 | 5 | 5 | - | - | - | - |
| Reka Core | 5 | 5 | 5 | 5 | 5 | 5 | 5 | 5 | - | - | - | - |
| Reka Flash | 5 | 5 | 5 | 5 | 5 | 5 | 5 | 5 | - | - | - | - |
| LLaMA 3.1 8b | 5 | 5 | 5 | 5 | 5 | 5 | 5 | 5 | - | - | - | - |
| LLaMA 3.1 70b | 5 | 5 | 5 | 5 | 5 | 5 | 5 | 5 | - | - | - | - |
| LLaMA 3.1 405b | 5 | 5 | 5 | 5 | 5 | 5 | 5 | 5 | - | - | - | - |
| Gemini 1.0 Pro | 5 | 5 | 5 | 5 | 5 | - | - | - | - | - | - | - |
| Mistral Large | 5 | 5 | 5 | 5 | 5 | - | - | - | - | - | - | - |
| Mistral Nemo | 5 | 5 | 5 | 5 | 5 | - | - | - | - | - | - | - |

Table 8: Number of repeats carried out for the Single Needle task.

| Model | Num. Repeats | | | | | | | | | | | |
|---|---|---|---|---|---|---|---|---|---|---|---|---|
| | 1.2k | 2.5k | 5k | 10k | 20k | 32k | 64k | 128k | 180k | 250k | 500k | 630k |
| Gemini 1.5 Pro | 5 | 5 | 5 | 5 | 5 | 5 | 5 | 5 | 1 | 1 | 1 | 1 |
| Gemini 1.5 Flash | 5 | 5 | 5 | 5 | 5 | 5 | 5 | 5 | 5 | 5 | 5 | 5 |
| Jamba 1.5 Large | 5 | 5 | 5 | 5 | 5 | 5 | 5 | 5 | - | - | - | - |
| Jamba 1.5 Mini | 5 | 5 | 5 | 5 | 5 | 5 | 5 | 5 | - | - | - | - |
| Claude 3.5 Sonnet | 5 | 5 | 5 | 5 | 5 | 5 | 5 | 5 | 5 | - | - | - |
| Claude 3 Sonnet | 5 | 5 | 5 | 5 | 5 | 5 | 5 | 5 | 5 | - | - | - |
| Claude 3 Haiku | 5 | 5 | 5 | 5 | 5 | 5 | 5 | 5 | 5 | - | - | - |
| GPT-4o | 5 | 5 | 5 | 5 | 5 | 5 | 5 | 5 | - | - | - | - |
| GPT-4o mini | 5 | 5 | 5 | 5 | 5 | 5 | 5 | 5 | - | - | - | - |
| Reka Core | 1 | 1 | 1 | 1 | 1 | 1 | 1 | 1 | - | - | - | - |
| Reka Flash | 1 | 1 | 1 | 1 | 1 | 1 | 1 | 1 | - | - | - | - |
| LLaMA 3.1 8b | 2 | 2 | 2 | 2 | 2 | 2 | 2 | 2 | - | - | - | - |
| LLaMA 3.1 70b | 2 | 2 | 2 | 2 | 2 | 2 | 2 | 2 | - | - | - | - |
| LLaMA 3.1 405b | 1 | 1 | 1 | 1 | 1 | 1 | 1 | 1 | - | - | - | - |
| Gemini 1.0 Pro | 5 | 5 | 5 | 5 | 5 | - | - | - | - | - | - | - |

Table 9: Number of repeats carried out for the Multiple Needles task.

| Model | Num. Repeats | | | | | | | | | | | |
|---|---|---|---|---|---|---|---|---|---|---|---|---|
| | 1.2k | 2.5k | 5k | 10k | 20k | 32k | 64k | 128k | 180k | 250k | 500k | 630k |
| Gemini 1.5 Pro | 5 | 5 | 5 | 5 | 5 | 5 | 5 | 5 | 1 | 1 | 1 | - |
| Gemini 1.5 Flash | 5 | 5 | 5 | 5 | 5 | 5 | 5 | 5 | 5 | 5 | 5 | 5 |
| Jamba 1.5 Large | 5 | 5 | 5 | 5 | 5 | 5 | 5 | 5 | - | - | - | - |
| Jamba 1.5 Mini | 5 | 5 | 5 | 5 | 5 | 5 | 5 | 5 | - | - | - | - |
| Claude 3.5 Sonnet | 5 | 5 | 5 | 5 | 5 | 5 | 5 | 5 | 5 | - | - | - |
| Claude 3 Sonnet | 5 | 5 | 5 | 5 | 5 | 5 | 5 | 5 | 5 | - | - | - |
| Claude 3 Haiku | 5 | 5 | 5 | 5 | 5 | 5 | 5 | 5 | 5 | - | - | - |
| GPT-4o | 5 | 5 | 5 | 5 | 5 | 5 | 5 | 5 | - | - | - | - |
| GPT-4o mini | 5 | 5 | 5 | 5 | 5 | 5 | 5 | 5 | - | - | - | - |
| Reka Core | 1 | 1 | 1 | 1 | 1 | 1 | 1 | - | - | - | - | - |
| Reka Flash | 1 | 1 | 1 | 1 | 1 | 1 | 1 | 1 | - | - | - | - |
| LLaMA 3.1 8b | 1 | 1 | 1 | 1 | 1 | 1 | 1 | 1 | - | - | - | - |
| LLaMA 3.1 70b | 1 | 1 | 1 | 1 | 1 | 1 | 1 | 1 | - | - | - | - |
| LLaMA 3.1 405b | 1 | 1 | 1 | 1 | 1 | 1 | 1 | 1 | - | - | - | - |
| Gemini 1.0 Pro | 5 | 5 | 5 | 5 | 5 | - | - | - | - | - | - | - |

Table 10: Number of repeats carried out for the Conditional Needles task.

| Model | Num. Repeats | | | | | | | | | | | |
|---|---|---|---|---|---|---|---|---|---|---|---|---|
| | 1.2k | 2.5k | 5k | 10k | 20k | 32k | 64k | 128k | 180k | 250k | 500k | 630k |
| Gemini 1.5 Pro | 5 | 5 | 5 | 5 | 5 | 5 | 5 | 5 | - | - | - | - |
| Gemini 1.5 Flash | 5 | 5 | 5 | 5 | 5 | 5 | 5 | 5 | 5 | 5 | 5 | 5 |
| Jamba 1.5 Large | 5 | 5 | 5 | 5 | 5 | 5 | 5 | 5 | - | - | - | - |
| Jamba 1.5 Mini | 5 | 5 | 5 | 5 | 5 | 5 | 5 | 5 | - | - | - | - |
| Claude 3.5 Sonnet | 5 | 5 | 5 | 5 | 5 | 5 | 5 | 5 | 5 | - | - | - |
| Claude 3 Sonnet | 5 | 5 | 5 | 5 | 5 | 5 | 5 | 5 | 5 | - | - | - |
| Claude 3 Haiku | 5 | 5 | 5 | 5 | 5 | 5 | 5 | 5 | 5 | - | - | - |
| GPT-4o | 5 | 5 | 5 | 5 | 5 | 5 | 5 | 5 | - | - | - | - |
| GPT-4o mini | 5 | 5 | 5 | 5 | 5 | 5 | 5 | 5 | - | - | - | - |
| Reka Core | 1 | 1 | 1 | 1 | 1 | 1 | 1 | - | - | - | - | - |
| Reka Flash | 1 | 1 | 1 | 1 | 1 | 1 | 1 | 1 | - | - | - | - |
| LLaMA 3.1 8b | 4 | 4 | 4 | 4 | 4 | 4 | 4 | 4 | - | - | - | - |
| LLaMA 3.1 70b | 3 | 3 | 3 | 3 | 3 | 3 | 3 | 2 | - | - | - | - |
| LLaMA 3.1 405b | 1 | 1 | 1 | 1 | 1 | 1 | 1 | 1 | - | - | - | - |
| Gemini 1.0 Pro | 5 | 5 | 5 | 5 | 5 | - | - | - | - | - | - | - |
| Mistral Large | 5 | 5 | 5 | 5 | 5 | - | - | - | - | - | - | - |
| Mistral Nemo | 5 | 5 | 5 | 5 | 5 | - | - | - | - | - | - | - |

Table 11: Number of repeats carried out for the Threading task.

| Model | Num. Repeats | | | | | | | | | | | |
|---|---|---|---|---|---|---|---|---|---|---|---|---|
| | 1.2k | 2.5k | 5k | 10k | 20k | 32k | 64k | 128k | 180k | 250k | 500k | 630k |
| Gemini 1.5 Pro | 1 | 1 | 1 | 1 | 1 | 1 | 1 | 1 | - | - | - | - |
| Gemini 1.5 Flash | 5 | 5 | 5 | 5 | 5 | 5 | 5 | 5 | 5 | 5 | 5 | 5 |
| Jamba 1.5 Large | 1 | 1 | 1 | 1 | 1 | 1 | 1 | 1 | - | - | - | - |
| Jamba 1.5 Mini | 1 | 1 | 1 | 1 | 1 | 1 | 1 | 1 | - | - | - | - |
| Claude 3.5 Sonnet | 5 | 5 | 5 | 5 | 5 | 5 | 5 | 5 | 1 | - | - | - |
| Claude 3 Sonnet | 1 | 1 | 1 | 1 | 1 | 1 | 1 | 1 | 1 | - | - | - |
| Claude 3 Haiku | 5 | 5 | 5 | 5 | 5 | 5 | 5 | 5 | 5 | - | - | - |
| GPT-4o | 1 | 1 | 1 | 1 | 1 | 1 | 1 | 1 | - | - | - | - |
| GPT-4o mini | 1 | 1 | 1 | 1 | 1 | 1 | 1 | 1 | - | - | - | - |
| Reka Core | 1 | 1 | 1 | 1 | 1 | 1 | 1 | - | - | - | - | - |
| Reka Flash | 1 | 1 | 1 | 1 | 1 | 1 | 1 | 1 | - | - | - | - |
| LLaMA 3.1 8b | 1 | 1 | 1 | 1 | 1 | 1 | 1 | 1 | - | - | - | - |
| LLaMA 3.1 70b | 1 | 1 | 1 | 1 | 1 | 1 | 1 | 1 | - | - | - | - |
| LLaMA 3.1 405b | 1 | 1 | 1 | 1 | 1 | 1 | 1 | 1 | - | - | - | - |
| Gemini 1.0 Pro | 5 | 5 | 5 | 5 | 5 | - | - | - | - | - | - | - |

Table 12: Number of repeats carried out for the Multi-threading task.

