# OpenReview forum: "Needle Threading: Can LLMs Follow Threads Through Near-Million-Scale Haystacks?"
_ICLR.cc/2025/Conference — ICLR 2025 Poster_

### Official Review · Reviewer_wcWb · 2024-10-24

**Soundness:** 3
**Presentation:** 4
**Contribution:** 3
**Rating:** 8
**Confidence:** 3

**Summary:**

This paper executed a set of experiments to evaluate the capabilities of 17 LLMs to perform retrieval tasks with long contexts. It introduces challenging multi-step threading and multi-threading retrieval tasks to test the models' ability to track information through extended contexts. The results show that increased context length reduces retrieval performance, with accuracy generally decreasing as the context window grows. The authors also demonstrate that many leading LLMs are thread-safe. Additionally, they highlight significant differences in token counts across different tokenizers.

**Strengths:**

(1) This paper studies an important aspect of LLMs -- their ability to manage long contexts -- which hasn't been fully explored yet.

(2) The paper is well-written.

(3) The task designs are comprehensive and insightful, especially the comparison between forward and backward threading.

(4) The experiments are extensive, and the results provide valuable insights into LLM performance with long contexts.

**Weaknesses:**

(1) Code and data are not currently shared.

(2) The experiments in the paper rely on abstract retrieval tasks using synthetic data. These tasks can not reflect the complexity found in real-world, domain-specific applications. In practice, data often includes ambiguous language, diverse formats, and contextually relevant nuances. Moreover, the experiment design cannot test the model's ability to understand semantics.

**Questions:**

Given that the experiments rely on synthetic, abstract tasks, how do you anticipate the performance trends observed in your study would translate to real-world, domain-specific applications with more complex and noisy data?

---

> ### Author Response · Authors · 2024-11-28
> **Thanks for reviewing**
>
> Thank you very much again for your review comments, Reviewer wcWb. Due to some overlapping and shared concerns, we provided our rebuttal above last week as an Official Comment to all reviewers. Hopefully, our additional experiments and responses have addressed your concerns, especially those regarding the use of abstract data and the availability of experimental data and code. Please let us know if there is anything else that needs clarifying.
>
> Note: our rebuttal figures can be found at the start of the Appendix (Page 14) of the updated rebuttal revision PDF.

---

> > ### Comment · Reviewer_wcWb · 2024-11-28
> >
> > Thanks!  I will keep my original positive rating.

---

### Official Review · Reviewer_bsZQ · 2024-10-31

**Soundness:** 3
**Presentation:** 3
**Contribution:** 3
**Rating:** 6
**Confidence:** 4

**Summary:**

This paper investigates the ability of Large Language Models (LLMs) to handle complex information retrieval and reasoning tasks across long contexts. The authors conduct a series of retrieval experiments with 17 leading LLMs to assess their capability to follow information threads within extensive context windows, revealing that while many models can manage multiple threads without significant performance loss, their effective context limits are often shorter than their technical limits. The study also emphasizes the variability in token counts between different tokenizers, which affects model performance comparisons. A key contribution is the introduction of a task-specific effective context limit metric, which provides a more nuanced understanding of model capabilities in long-context scenarios.

**Strengths:**

1. This paper provides an extensive evaluation of 17 leading LLMs across various long-context retrieval tasks, offering a thorough analysis of their performance in handling complex information retrieval and reasoning.

2. Through innovative needle threading and multi-threading experiments, the study creates scenarios where LLMs must follow chains of information across different parts of the context, effectively testing their limits in long-context understanding.

3. Analysis reveals significant differences in tokenization between models, crucial for understanding the discrepancies in reported context lengths and making accurate comparisons between LLMs.

4. The authors propose a task-specific and configurable metric independent of tokenization, enabling more precise assessment of models' reasoning capabilities over context.

**Weaknesses:**

1. The study's exclusive use of synthetic data (UUID key-value pairs) may not accurately reflect performance on natural language tasks or domain-specific applications.

2. The paper may be limited in techinical contribution.

**Questions:**

1. Have you considered conducting smaller-scale validation experiments with natural language data to verify if the findings generalize?

2. For the branched threading task, why was it only evaluated on a subset of models and smaller context sizes?

---

> ### Author Response · Authors · 2024-11-28
> **Thanks for reviewing**
>
> Thank you very much again for your review comments, Reviewer bsZQ. Due to some overlapping and shared concerns, we provided our rebuttal above last week as an Official Comment to all reviewers. Hopefully, our additional experiments and responses have addressed your concerns, especially those regarding the use of abstract data and expanding the scope of the branched threading experiment. Please let us know if there is anything else that needs clarifying.
>
> Note: our rebuttal figures can be found at the start of the Appendix (Page 14) of the updated rebuttal revision PDF.

---

> ### Comment · Reviewer_bsZQ · 2024-11-29
>
> Thanks for the authors' responses. I will keep my original score.

---

### Official Review · Reviewer_n6xp · 2024-11-08

**Soundness:** 3
**Presentation:** 2
**Contribution:** 3
**Rating:** 5
**Confidence:** 2

**Summary:**

The paper evaluates the performance of Large Language Models (LLMs) as their context limits increase, focusing on their ability to handle complex information retrieval across multiple documents. Through experiments with 17 leading LLMs, the study finds that while many models effectively manage multiple threads of information, their actual efficient context range is often shorter than their maximum allowed, leading to reduced performance as the context window expands. The research also highlights inconsistencies in token counts across different tokenizers. Results and methodologies are made available for further study.

**Strengths:**

Note: This is a review by an emergency reviewer.

The paper addresses an intriguing and valuable research problem by exploring the utility of synthetic data for abstract tasks in the context of long-sequence processing.

Introducing multi-threading tasks as part of the experimental setup is particularly innovative.

The findings of the paper offer practical insights that could inform both the academic and industry sectors about the design and tuning of language models for specific applications.

**Weaknesses:**

Note: This is a review by an emergency reviewer.

The major concern is that there's a gap between experimental design and real-world applications. The study employs highly abstract tasks using synthetic data with no natural language semantics, deviating considerably from the typical environments where large language models (LLMs) operate. The string-serialized JSON objects and UUIDs as key-value pairs fail to engage the models in natural language processing—core to their training and operational objectives. Consequently, the findings have limited applicability to real-world scenarios that demand comprehension, generation, and manipulation of actual linguistic content. This gap undermines the relevance of the research to practical applications of LLMs, which are primarily designed to interact with and generate coherent, contextually appropriate natural language.

The authors mention that they have released their code and experimental data for public use. However, the authors didn't upload supplemental materials to the open review nor include an anonymous link of their data and code. I would suggest sharing the code through an anonymous GitHub repository or similar platform. This would greatly aid other researchers and reviewers in replicating and understanding the research.

**Questions:**

Please refer to the previous section

---

> ### Author Response · Authors · 2024-11-28
> **Thanks for reviewing**
>
> Thank you very much again for your review comments, Reviewer n6xp. Due to some overlapping and shared concerns, we provided our rebuttal above last week as an Official Comment to all reviewers. Hopefully, our additional experiments and responses have addressed your concerns, especially those regarding the use of abstract data and the availability of experimental data and code. Please let us know if there is anything else that needs clarifying.
>
> Note: our rebuttal figures can be found at the start of the Appendix (Page 14) of the updated rebuttal revision PDF.

---

> ### Comment · Reviewer_n6xp · 2024-11-29
>
> Thank you for your detailed response.
>
> I appreciate that you have provided the dataset in the rebuttal period, addressing half of my initial concerns.

---

> > ### Author Response · Authors · 2024-11-30
> >
> > Thank you for your response and for acknowledging our efforts to provide the dataset and code. We appreciate your feedback. Regarding your other concerns, could you please let us know if the additional natural language experiments and discussion in the **Use of abstract/synthetic data & applicability to real-world natural language scenarios (Q7CU, n6xp, bsZQ, wcWB)** section in our rebuttal have addressed them or if there's anything else we can clarify?

---

### Official Review · Reviewer_Q7CU · 2024-11-14

**Soundness:** 3
**Presentation:** 3
**Contribution:** 3
**Rating:** 6
**Confidence:** 4

**Summary:**

In this paper, the authors introduce simple single-needle retrieval, multiple-needle and conditional-needle retrieval and challenging needle threading and multithreading retrieval. Experiments on haystacks consisting of key-value pairs of UUIDs shows the retrieval precisions of 17 LLMs vary on different context lengths, multiple-needle and multiple threading conditions.

**Strengths:**

The paper focuses on an interesting problem: how effectively LLMs use their context.
The observations from the experiments are inspiring, for example, many models are remarkably thread-safe: capable of simultaneously following multiple threads without significant loss.

**Weaknesses:**

Only one synthetic dataset is used for evaluation.

**Questions:**

What is the time complexity (running/response time) of each LLM that is used for evaluation in different experiment settings?

---

> ### Author Response · Authors · 2024-11-28
> **Thanks for reviewing**
>
> Thank you very much again for your review comments, Reviewer Q7CU. Due to some overlapping and shared concerns, we provided our rebuttal above last week as an Official Comment to all reviewers. Hopefully, our additional experiments and responses have addressed your concerns, especially those regarding the use of abstract data and time complexity. Please let us know if there is anything else that needs clarifying.
>
> Note: our rebuttal figures can be found at the start of the Appendix (Page 14) of the updated rebuttal revision PDF.

---

### Author Response · Authors · 2024-11-23
**Initial response to reviewers**

We thank the reviewers for their insightful comments, positive responses, and time spent reviewing our paper. We are glad the reviewers found our paper to focus on an interesting (Q7CU) and intriguing (n6xp) problem with innovative experiments (bsZQ) and task designs that are comprehensive and insightful (wcWb).

Here, we address the weaknesses and questions raised in your reviews. **EDIT**: We have uploaded a Rebuttal Revision PDF of our paper with an additional ‘Rebuttal Figures’ section at the start of the Appendix (Page 14) containing some figures related to our rebuttal. Note, for the final version, we will incorporate any changes into the main paper and appendix, ensuring the total length is within the 10-page limit.



**Use of abstract/synthetic data & applicability to real-world natural language scenarios (Q7CU, n6xp, bsZQ, wcWB):**

Thank you for raising this important point. As mentioned in our paper, focusing our evaluation on abstract and synthetically generated scenarios has benefits of enabling fine-grained control and high-quality noise-free generation without potentially expensive annotation. Additionally, the likelihood of data contamination is significantly reduced and we can be confident – especially given the unique nature of UUIDs – that our tasks cannot be performed via memorisation.

However, we concur that there is a gap between our experimental design and real-world applications. To address this, we have carried out a series of natural-language experiments. We focus on the threading and multi-threading tasks as these arguably had the most interesting takeaways. Concretely, we took The History of the Decline and Fall of the Roman Empire book, by Edward Gibbon (see Fig. 1 of our paper) as our “distractor” text. We chose a sentence in the text as a proxy for a UUID pair in the abstract task. Then, we prompted OpenAI’s o1-preview model to provide a list of plausible yet fictional Roman events (that aren’t included in the text). Using these events, we constructed ‘threads’ in the form of linked sentences:
Sentence 1: ‘Event A and then Event B.’,...
Sentence 2: ‘Event B and then Event C.’,... and so on.
And replaced sentences in the text with these thread sentences.

Thus, we have a natural language text containing linked information that is *in distribution* with the rest of the text. This serves as a natural language proxy for our abstract threading tasks.

*Single Threading results:*

We repeated the threading task in this new setting by evaluating models on haystacks from 1k to 630k token context lengths with threads from 2-25 steps in length. Please refer to Fig. 14 in the rebuttal revision pdf for selected heatmaps for threads travelling in the forward and backward directions. As in the abstract setting (Fig. 10), following threads in the natural language text proves challenging for the models, with similar poorer performance observed at longer contexts. The preference towards forward travelling threads is clearly also apparent from the heatmaps in this setting. We will update the final paper with these observations.

*Multi-threading results:*

We conducted multi-threading experiments in the natural language setting using the same approach as the Single Threading experiments above (this time with additional simultaneous threads). We present results for the Thread-Safe experiments repeated in the natural language setting in Figs. 15 and 16 in the rebuttal revision pdf. Each point represents an average over 5 repeats retrieving threads with a length of 3 in 20k LLaMA 3.1 token haystacks. For Fig. 15, threads are randomly-directed whereas for Fig. 16 they are all forward-directed. Unlike the Single Threading experiments – for which the results and insights were largely the same across the abstract and natural text settings – the Multi-threading task in the natural language setting proved much more challenging for the models. Moreover, we find the task to be challenging when retrieving multiple threads that are all forward or all randomly directed. Similar results were obtained when repeating this experiment on smaller haystacks and chain lengths of 2. Thus, our study reveals that the multi-threading results are nuanced – with strong performance in the synthetic setting and weaker performance in the natural text setting. We’ll update the paper with these new insights, which we hope will be valuable for the research community.

---

> ### Author Response · Authors · 2024-11-23
> **Initial response to reviews (Pt. 2)**
>
> **Anonymous Links for Code and Data (n6xp, wcWb):**
>
> Thank you for raising this point. Our data and code are available through a HuggingFace and GitHub repository, respectively, via a project page. This will be released in the final version, however to maintain anonymity, we did not include the links in our submitted version. For use during review, here is a link to an anonymous HuggingFace repository containing our experimental data and some example inference code: https://huggingface.co/datasets/needle-threading/needle-threading.
>
>
> **Time complexity/Response time (Q7CU):**
>
> Good suggestion, thanks. We plot the mean response times for a selection of models from our two natural language experimental settings in the rebuttal revision pdf. In Fig. 17 we plot the response times for the threading experiment. The results show an overall trend of increasing response time at longer context lengths, as well as, variation in the response times based on provider/model. In Fig. 18 we plot the response times for our multi-threading experiment. The mean response time increases with the number of threads to retrieve, as the models have to generate output tokens corresponding to the event at the end of each additional thread. These results offer insights into the current expected response times for different models and experimental settings but it is worth caveating their dependence on the hardware serving the models via an API at the time of inference.
>
>
> **Additional branched threading results (bsZQ):**
>
> We have begun running the branched threading experiments at additional context lengths and models. We will present results from this before the rebuttal period ends.

---

> > ### Author Response · Authors · 2024-11-28
> > **Branched threading results**
> >
> > We have now evaluated the branched threading experiment with additional models and longer context lengths. We have updated the relevant figure (Fig. 14) in the Rebuttal Revision PDF with these additional results. There are a very small number of the longest context lengths that we have yet to evaluate for some models, though we expect they will follow a similar trend. We will ensure these few remaining evaluations are completed and included in the final version.

---

### Author Response · Authors · 2024-12-04

We thank the reviewers again for their time, helpful comments and positive responses to our paper. In our rebuttal, we aimed to address all your concerns and consider our paper to be improved as a result.

---

### Meta-Review · Area_Chair_TAZy · 2024-12-21

**Metareview:**

This paper focuses on experimental validation and discussion of the context understanding capabilities of existing LLMs. The authors designed a series of retrieval experiments, including single needles, and multiple needles with random and clustered distributions. The experimental process is clearly presented and easy to understand, and most reviewers expressed their recognition of this work and gave positive scores. The main concern raised by the reviewers was that the authors only used synthetic datasets for testing, which the authors addressed in the rebuttal phase. I personally find this acceptable, especially since the authors conducted thorough experiments and provided a complete experimental process. Additionally, my suggestion is for the authors to avoid using different x-axis intervals (e.g., Gemini 1.5 Flash and GPT-4 Mini) in the same figure (e.g., Figure 5). To summarize, I recommend accepting this work.

**Additional Comments On Reviewer Discussion:**

The authors addressed the issues raised by the reviewers during the rebuttal phase.

---

### Decision · Program_Chairs · 2025-01-22

Accept (Poster)